



# Radiative and climate effects of aerosol scattering in long-wave radiation based on global climate modeling

Thomas Drugé[1], Pierre Nabat[1], Martine Michou[1], and Marc Mallet[1]

[1]CNRM, Université de Toulouse, Météo-France, CNRS, Toulouse, France

**Correspondence:** Thomas Drugé (thomas.druge@meteo.fr)

**Abstract.** The few studies that considered aerosol scattering in the long-wave (LW) typically relied on artificially increasing it. In order to analyze the radiative and climatic effects of physically accounting for this process, simulations have been performed with the ARPEGE-Climat atmospheric global climate model over the 1985-2014 period, using the ecRad radiation scheme, and updated optical properties of coarse aerosols, particularly dust. The evaluation of the model coarse aerosol optical depth

(AOD) against AERONET data over North Africa and Arabian Peninsula shows the ability of ARPEGE-Climat to capture spatio-temporal variations of coarse AOD, despite regional biases. The comparison of simulations with and without aerosol scattering in the LW shows that this process leads to a significant increase in downwelling surface LW radiation in dust-emitting regions (+5 W m$^{-2}$ on average) between March and September, in line with the maximum coarse AOD. This increase results in a rise in minimum near-surface temperatures of up to +1°C. It is also associated with an outgoing LW radiation decrease at

the top of the atmosphere (TOA). However, during certain months and regions, near-surface temperatures can be significantly reduced due to short-wave surface radiation decreases related to increases in low-level clouds. A precipitation increase over Sahel during September linked to wetter atmospheric layers is also simulated. Neglecting LW aerosol scattering in climate simulations has therefore significant impacts on climate, notably in dust-emitting regions. Globally, the LW aerosol scattering contribution to radiation is of 0.4 W m$^{-2}$ at both surface and TOA.

## 1 Introduction

Aerosols impact the climate by disrupting the Earth's energy budget. The latest Intergovernmental Panel on Climate Change Assessment Report (Masson-Delmotte et al., 2021) highlights that the dominant contribution to the aerosol Effective Radiative Forcing (ERF) arises from aerosol-cloud interactions (ERFaci), with high confidence. The 1750-2014 ERFaci is assessed to be –1.0 [–1.7 to –0.3] W m$^{-2}$ (medium confidence), while the other part of the ERF, attributed to aerosol–radiation interactions

(ERFari), is assessed to be –0.3 [–0.6 to 0.0] W m$^{-2}$ (medium confidence). Regarding aerosol-radiation interactions, which are of particular interest to this paper, a deeper understanding of the processes governing aerosol radiative properties has emerged since the previous IPCC report (AR5). The magnitude of ERFari in the AR6 has been reduced by about 50% compared to the AR5, based on agreement between observation-based and modelling-based evidence. A synthesis of the literature cited in (Masson-Delmotte et al., 2021) is that short-wave (SW) flux changes can be attributed to aerosol–radiation and aerosol–cloud

interactions, while the small positive long-wave (LW) flux changes are associated with aerosol–cloud interactions, particularly



linked to changes in liquid-water path. However, LW flux changes resulting from aerosol-radiation interactions are not mentioned even though uncertainties remain, particularly concerning assumptions about aerosol emissions masses, size distribution, optical properties (Hess et al., 1998; Dobbie et al., 2003) and mixing states (Myhre et al., 2013; Szopa et al., 2021).

Several studies have shown that, even though the effects of most aerosol species (particularly fine particles) on LW radiation are small compared to their effects on SW radiation, highly absorbing and scattering large particles, such as mineral dust, have a LW forcing that can counteract their cooling effect in the SW (Fouquart et al., 1987; Hansell et al., 2010; Di Sarra et al., 2011; Sicard et al., 2014; Di Biagio et al., 2020). Additionally, a growing body of recent work focusing on the microphysical properties and radiative effects of dust in the LW range is now available (Hansell et al., 2010; Haywood et al., 2011; Köhler et al., 2011; Osborne et al., 2011; Weinzierl et al., 2011; Sicard et al., 2014; Di Biagio et al., 2017, 2019, 2020; Fountoulakis et al., 2024). Often cited in the subject, Dufresne et al. (2002), using a radiative transfer model and standard vertical profiles of dust aerosol, highlight the importance of the mineral aerosol scattering on LW radiation. The study shows that neglecting this effect may lead to an underestimate of the LW aerosol forcing of about 50% at the top of the atmosphere (TOA) and 15% at the surface. Both Dufresne et al. (2002) and Sicard et al. (2014) also found that the LW aerosol RF is maximum at wavelengths between 8 and 13 μm, while Sicard et al. (2014) indicated that large particles have a non-negligible effect in the 17 to 22 μm range at the TOA. Furthermore, Dufresne et al. (2002) demonstrated that the LW aerosol scattering only slightly affects heating rates inside the atmosphere: neglecting it leads to a maximum reduction of 10% in the cooling caused by aerosols at the top of the aerosol layer, while slightly increasing the cooling at the surface.

Despite these studies, the aerosol LW radiative forcing is still only partially accounted for in climate models, either global or regional. While the aerosol LW absorption is considered, LW scattering is not (Sicard et al., 2014; Granados-Muñoz et al., 2019; Di Biagio et al., 2020). In the best cases, the missing aerosol LW scattering is "artificially" accounted for by increasing either the aerosol optical depth (AOD) or the retrieved TOA direct radiative effect (DRE) using constant correction factors that are independent from the dust situation. For example, in Miller et al. (2006), LW scattering is represented by a 30% increase in dust optical thickness, based on the calculations of Dufresne et al. (2002). In their study Kok et al. (2017) follow this conservative approach, assuming that LW scattering enhances the LW absorption radiative effect by a factor of 0.3, accounting for 23% of the LW DRE at the TOA. However, these corrections represent only about half the value estimated by Dufresne et al. (2002) and Sicard et al. (2014). A different example is demonstrated by Di Biagio et al. (2020) who corrected their LW DRE, calculated as the difference in LW radiative fluxes with and without dust in the LMDZOR-INCA model, by applying a multiplicative factor of 2.04 at the TOA (where the scattering contribution to the TOA LW DRE is then 51%) and of 1.18 at the surface (where the scattering contribution to the surface LW DRE is then 15%), based on Dufresne et al. (2002). Di Biagio et al. (2020) estimated a global annual mean all-sky LW DRE of mineral dust at TOA of $+0.22$ W m$^{-2}$ which lies between the AEROCOM median estimate ($+0.15$ W m$^{-2}$) and the estimate of (Kok et al., 2017) ($+0.29$ W m$^{-2}$). Similarly, because their radiative transfer model did not account for aerosol LW scattering, Ito et al. (2021) followed Di Biagio et al. (2020) and multiplied their LW radiative fluxes by the adjustment factors from Di Biagio et al. (2020). Likewise, Li et al. (2021) artificially increased the LW dust DRE at the TOA by 51% to account for scattering effects neglected by the global atmosphere model CAM. With this adjustment, they estimated a LW dust DRE at the TOA between $+0.14$ and $+0.20$ W m$^{-2}$. A final example is provided by Hogan and Bozzo



(2018) who ran the European Centre for Medium-Range Weather Forecasts (ECMWF) Numerical Weather predictions (NWP) model with the ecRad radiation scheme able to consider the LW aerosol scattering (see also paragraph 2.1). They concluded that turning on aerosol LW scattering impacts global mean surface and TOA LW irradiances only up to +0.1 W m$^{-2}$, which is far from the 3 to 5 W m$^{-2}$ cited in Dufresne et al. (2002), and thus negligible in the context of NWP.

In addition to the inadequate representation of aerosol LW scattering, global and regional climate models also struggle with accurately representing various characteristics of coarse particles, particularly dust. This limitation is especially important for the LW DRE, as these coarse particles have the greatest impact in this spectral range (Dufresne et al., 2002; Di Biagio et al., 2020). Sicard et al. (2014) demonstrated that LW scattering has no effect on aerosol forcing for radii lower than 0.1 μm. However, for particles with radii larger than 0.1 μm they estimated that this process contributes up to 38% of the LW aerosol

forcing at the TOA and up to 18% at the surface, with the highest contribution coming from particles with a radius of 0.5 μm. In their study, Di Biagio et al. (2020) used a superposition of four lognormal modes to represent the aerosol size distribution in their aerosol model. They showed that the mode with a mass median diameter (MMD) of 7 μm represents more than 60% of the DRE LW. Additionally, they showed that the fraction of dust with a MMD above 20 μm contributes to about 30% of the DRE. Climate models, however, tend to overestimate the mass concentration of dust particles with a diameter smaller than 2 μm, and

underestimate the concentration of large dust particles (greater than 5 μm) compared to observations (Kok, 2011; Kok et al., 2017; Ryder et al., 2019; Di Biagio et al., 2020). Kok et al. (2017) applied constraints on dust emission sizes to better match observations and concluded that particles with diameters smaller than 20 μm contribute an average of 4.3% of the emitted dust mass, which is significantly lower than the 5–35% assumed in many global models. Furthermore, Van Der Does et al. (2018) highlighted new observations suggesting the presence of giant mineral dust particles, with a diameter larger than 75 μm and

up to 450 μm, far from their source, 2400 and 3500 km away. Another challenge is the difficulty in accurately quantifying the optical properties of dust in the LW spectral range, despite recent advances in this area (Granados-Muñoz et al., 2019). Most global models use the dust complex refractive index described in Volz (1973) which is based on dust collected at Barbados after being transported from the Sahara. However, recent laboratory measurements of dust samples suggest that the imaginary part of this refractive index is too high, which could lead to an overestimation of dust absorption (Di Biagio et al., 2014, 2017, 2019).

Finally, dust particles are generally considered spherical in climate models, this only has a limited impact on the DRE at the TOA (Bellouin et al., 2004; Colarco et al., 2014).

    In this study, we analyze the radiative and climatic impacts of the aerosol scattering in the LW spectral range through simulations using a new version of the CNRM (National Centre for Meteorological Research) global climate atmospheric model, ARPEGE-Climat. Unlike previous studies, we parameterize aerosol scattering in the LW spectrum within the radiation

scheme ecRad (Hogan and Bozzo, 2018), which is implemented in ARPEGE-Climat. This approach allows us to estimate the impact of LW aerosol scattering through its physical representation. This work also involved updating the optical properties of aerosols in the LW spectrum, particularly those of dust. Sect. 2 describes the climate model and the simulations carried out, while the updated aerosol optical properties are detailed in Sect. 3. Sect. 4 presents the evaluation of coarse AOD, with a particular focus on the local scale, through comparison with Aerosol Robotic Network (AERONET) data. Sect. 5 analyzes the

model results, and Sect. 6 summarizes the conclusions.



## 2 Methodology

### 2.1 The ARPEGE-Climat global climate model

The present study has been conducted with a new version of the ARPEGE-Climat global atmosphere model which is the atmospheric component of the Centre National de Recherches Météorologiques (CNRM) climate model. This new version, referred to as v7.0.1, is an update of version 6.3 (Roehrig et al., 2020) which was used for the sixth phase of the Coupled Model Intercomparison Project (CMIP6). V7.0.1 is based on cycle 48t1_op1 of the ARPEGE/IFS system, a system developed jointly by Météo-France and the ECMWF for both NWP and climate applications.

As other atmospheric models, ARPEGE-Climat consists in a dry dynamical core and a suite of physical parameterizations that represent diabatic processes. In short, ARPEGE-Climat v7.0.1 very largely shares the choices made for the ARPEGE NWP version which means that these choices differ from those of the ARPEGE-Climat CMIP6 version. This is the case in particular for the parameterisations of the deep convection and of the radiation. Specifically, the convection scheme is based on the work of (Tiedtke, 1989), with many subsequent modifications, as reported in (Bechtold et al., 2008, 2014; Becker et al., 2021). The radiation scheme used is "ecRad" (Hogan and Bozzo, 2018), which has been operational in the IFS NWP model since 2017. The ecRad scheme represents the latest advancement in several decades of development to improve the radiative transfer scheme used in the IFS. It provides multiple options for handling sub-grid cloud structure and the optical properties of gases, aerosols, and clouds. The configuration used in this study computes gas optical properties using the Rapid Radiation Transfer Model for GCMs, RRTMG, (Mlawer et al., 1997; Iacono et al., 2008) across sixteen spectral bands in the LW and fourteen in the SW. It also treats the cloud sub-grid structure using the Monte Carlo Independent Column Approximation (McICA, Pincus et al. (2003)). The long-wave scattering of clouds and/or aerosols can be turned on or off. We kept LW scattering of clouds on in all our simulations, and turned on or off the LW scattering of the aerosols (see paragraph 2.2).

The radiation scheme in ARPEGE-Climat is provided with profiles of aerosol mass mixing ratios for various aerosol species. During the setup phase of ecRad, each aerosol species is mapped to pre-computed aerosol optical properties, which are stored in a netCDF file. These properties are calculated using Mie theory and averaged to the RRTMG bands. For each aerosol species, the optical properties include the mass-extinction coefficient, the single scattering albedo and the asymmetry factor. Some aerosol species are hydrophilic, meaning that their optical properties are stored as a function of relative humidity, with bins of 10% width (except between 80 and 100% where the bin width is 5%). When optical properties are computed for these species, the nearest bin is selected based on the current relative humidity.

Aerosol mass mixing ratios come from the TACTIC aerosol scheme, originally described by (Michou et al., 2015; Nabat et al., 2015a). This version has been largely evaluated through CMIP6 simulations (Michou et al., 2020) and further developed to include nitrate aerosols (Drugé et al., 2019) and brown carbon aerosols (Drugé et al., 2022). In the present study, seven aerosol types have been considered, excluding brown carbon. These include: desert dust, with three size bins (diameter limits of 0.01, 1.0, 2.5, and 20 μm), sea-salt with three size bins (diameter limits of 0.01, 1.0, 10.0, and 100.0 μm), sulfate in one bin, organic matter with two bins (hydrophilic and hydrophobic), black carbon with two bins (hydrophilic and hydrophobic), nitrate with two bins (formed by gas-to-particle reactions and heterogeneous chemistry) and ammonium in one bin. These



aerosol types represent the main anthropogenic and natural aerosol species of the troposphere, that we assume to be externally mixed. ARPEGE-Climat v7.0.1 accounts for interactions between particles and radiation (direct aerosol effect). However, the interactions between aerosols and clouds (indirect aerosol effects), are not considered in this version of the model. During ARPEGE-Climat simulations, aerosol optical properties are based on look-up tables pre-calculated using a Mie code and the aerosol sphericity hypothesis (Ackerman and Toon, 1981). These optical properties are also dependent on the relative humidity,

except for desert dust and hydrophobic black carbon and organic matter.

The atmospheric dynamics and physics are computed on a T127 triangular grid truncation or associated reduced gaussian grid, which corresponds to a spatial resolution of approximately 150 km in both longitude and latitude. ARPEGE-Climat is a "high-top" model with 91 vertical levels, extending from the surface to 0.01 hPa in the mesosphere. These levels use hybrid $\sigma$-pressure coordinates (Simmons and Burridge, 1981), with 15 levels below 1500 m. To simulate surface state variables

and fluxes, ARPEGE-Climat uses the SURFace EXternalisée (SURFEX) modeling platform, in its version 8.1 here which is present in the cycle 48t1 of the ARPEGE/IFS system (Le Moigne et al., 2020). This platform operates over the same grid and with the same time step as the rest of the model. Physical processes at the land surface are represented using the Interaction Soil-Biosphere-Atmosphere (ISBA) land surface model (Noilhan and Mahfouf, 1996).

## 2.2 Model simulation configurations and reference dataset

Two configurations of ARPEGE-Climat, specifically of the ecRad radiation scheme, have been used in this study, turning on or off the LW aerosol scattering, to run two amip-type simulations over a 30-year period (1985-2014). In the rest of the paper we refer these two simulations as NOLWAS, without, and LWAS, with LW aerosol scattering. The forcings used in these simulations are those from the CMIP6 framework (Eyring et al., 2016), which include, among others, sea surface temperatures and sea-ice concentrations. As for the aerosol forcing, to minimize computational time, we have used a 3D monthly climatology

of aerosol concentrations (14 aerosol bins or species in total) over the ARPEGE-Climat grid that we provided as input to ecRad in the NOLWAS and LWAS simulations. This climatology is based on a 10-year simulation (2005-2014) with the version of the TACTIC aerosol scheme described in Drugé et al. (2022), without brown carbon as it was not relevant for this study.

We used data from the Aerosol Robotic Network (AERONET) (Holben et al., 1998), which are available for the period 2000-2020 depending on the station, to evaluate the ability of ARPEGE-Climat to reproduce coarse AOD. AERONET is a

globally distributed network of ground-based sun photometers that provide local, column-integrated aerosol properties, including total, coarse and fine AOD at various wavelengths. For this study, we used version 3 data (level 1.5 with an automated cloud-screening) (Sinyuk et al., 2020). AOD at 550 nm was derived using Ångström coefficients between the closest available upper and lower wavelengths. Monthly climatologies were then computed over the available periods to compare with the model monthly climatology. It is worth mentioning that AERONET AOD measurements have an uncertainty less than 0.01 for

wavelengths longer than 440 nm and less than 0.02 for UV wavelengths (Eck et al., 1999; Kinne et al., 2013). Additionally, AERONET AOD are derived during daytime whereas our model AOD is averaged over day and night.

For the evaluation, we selected twelve AERONET stations that have enough data to compute an averaged annual cycle (one value per month) representative of the climate of the 2000-2020 period. In that sense a given month of a station can be retained





if it complies with the following rule: the 2000-2020 average can be calculated from monthly averages from three different years, each with at least 8 days of available measurements in the month. These stations, that we numbered in alphabetical order, are located in southern Europe, northern Africa and over the Arabian Peninsula, where the maximum amount of coarse AOD is found. In order to represent a region with significant sea-salt aerosols, we also include one station (station 1) in the southern Indian Ocean even if the observations available at this station do not meet our selection criteria. See Table A1 for characteristics of the AERONET stations of this study.

**3  Dust and sea-salt optical properties**

This section focuses on the desert dust and sea-salt aerosols, as these species are the main sources of coarse aerosols. We calculated optical properties from refractive indices (RI) using a Mie code.

For desert dust aerosols, a new RI has been introduced in TACTIC for this study, represented by purple dots in Figure 1, with the original version shown by red dots. This updated RI is based on several studies: Di Biagio et al. (2017) for wavelengths
comprised between 3 and 15 μm, and Woodward (2001) for higher wavelengths, between 15 and 40 μm. The study of Di Biagio et al. (2017) was selected because it presents a desert dust RI derived from the mineralogical composition and size distribution of mineral dust obtained through in situ measurements in a smog chamber. Their study covers 137 desert dust samples coming from various natural soils across eight regions including northern Africa, the Sahel, the Middle East, South America, and others. This large sample captures the diversity of sources and the heterogeneity of soil composition on the global scale. The
complex desert dust RI is finally derived through optical inversion using extinction spectrum and size distribution measured in the smog chamber. Di Biagio et al. (2017) observed significant variability in the imaginary LW RI across samples, ranging from 0.001 to 0.92, reflecting differences in particle composition. In contrast, the real part of the desert dust RI was less variable, ranging from 0.84 to 1.94. In our study, since the aerosol optical properties are not calculated on-line but based on pre-calculated look-up tables, an average value per wavelength between 3 and 15 μm was calculated from the Di Biagio et al.
(2017) study for both the real and imaginary parts of the desert dust RI. For wavelengths between 15 and 40 μm, as in the IFS model, our desert dust RI is based on an earlier study (Woodward, 2001), which provides desert dust RI values for both the real and imaginary parts derived from a range of measurements taken at various locations (Carlson and Benjamin, 1980; Sokolik et al., 1993, 1998).

As shown in Figure 1, the desert dust RI used in this study ranges from 1.2 to 2.1 for the real part and from 0 to 0.65 for
the imaginary part. Compared to OPAC (Hess et al., 1998), Krekov (Krekov, 1993) and IFS (Fouquart option; Fouquart et al. 1987), our desert dust RI shows some differences. Specifically, for its real part, Figure 1 indicates that the values used in this study are lower than those from Krekov and OPAC for wavelengths above 20 μm. Regarding the imaginary part, our values are lower than those from OPAC, IFS and Krekov. Compared to our initial values (red dots), our new imaginary desert dust RI shows higher values from 15 μm, bringing them closer to other datasets. Desert dust optical properties calculated from this
updated RI (not shown here) are now closer to those used in the IFS model.



The sea-salt RI used in this work remains unchanged from previous versions of the TACTIC scheme. It is based on the study by (Krekov, 1993), which presents values for 0% relative humidity. It is important to note that sea-salt RI has been less extensively studied than desert dust RI in the scientific literature. The real and imaginary parts of the sea-salt RI for several relative humidities (0, 50 and 80%) are shown in Figure 1. This figure illustrates that the sea-salt RI used in IFS, which

corresponds to OPAC data (Hess et al., 1998), is quite similar to the one used here. Similarly, (Irshad et al., 2009) report sea-salt real and imaginary RI values also consistent to those used here. However, for the imaginary part of the RI, the values at 0% relative humidity in this study are higher, between 10 and 30 μm, compared to those used in the IFS. Above 30 μm the values are lower. The optical properties calculated from this RI (not shown here) are similar to those used in the IFS model, except for the largest particles, which exhibit notably lower extinction and a higher asymmetry parameter. These differences are primarily

due to difference in the particle size between the coarsest bin used in the ARPEGE-Climat model and the one used in the IFS model.

## 4 Evaluation of the AOD of coarse aerosols

In the ARPEGE-Climat global climate model, coarse aerosols correspond to desert dust bins 2 (1-2.5 μm diameter) and 3 (2.5-20 μm diameter), sea-salt bins 2 (1-10 μm diameter) and 3 (10-100 μm diameter), as well as coarse nitrate (see Drugé et al.

(2019) for details). Figure 2 presents the annual mean of the coarse AOD simulated by ARPEGE-Climat over the period 1985-2014. The maximum coarse AOD, reaching values of around 0.40, is located in the Arabian Peninsula and in northern Africa (a region that will be referred as "N-Africa" in the rest of the paper), within the latitude band of [10°N - 30°N]. This maximum coarse AOD is consistent with the AeroCom phase 3 ensemble median (14 models) presented in Gliß et al. (2021). Over these regions, the seasonal cycle of coarse aerosol AOD is compared with measurements from various AERONET stations. The data

from stations in Niger (station numbered 2), Senegal (4), Saudi Arabia (11) and Algeria (12), are shown in Figure 2, and are representative of the model behavior in these regions. These graphs clearly demonstrate that desert dust largely contributes to coarse AOD over N-Africa and the Arabian Peninsula, with a peak occurring between March and September.

Correlation coefficients (r) between ARPEGE-Climat model results and AERONET coarse AOD measurements (climatological series) in dust-emitting regions are generally high (r > 0.61), indicating that the model effectively captures the spatio-

temporal variations in coarse AOD. The coarse AOD biases range from -0.11 to +0.10, depending on the location of the station. Specifically, stations in the northern part of the regions N-Africa and Arabian Peninsula, such as in Saudi Arabia (11) and Algeria (12), show a peak in coarse AOD during May and June, which is rather well reproduced by the model. However, measurements at these stations also reveal an overestimation of about 30 to 40 % (+0.10) in coarse AOD by the model throughout the year. In contrast, stations in the southern part of the region N-Africa - Arabian Peninsula, such as Niger (2) and Senegal

(4), show a significant underestimation of the coarse AOD simulated by the model with biases of about 40 to 50 % (-0.10) relative to AERONET data. In particular, the AERONET measurements at Niger (2) show a peak during March, April, May and June, which is not captured by the ARPEGE-Climat model. Similarly, stations in Cape Verde (3) and Mali (6), shown in Figure A1, also display an underestimation of coarse AOD simulated by the model. In comparison, Gliß et al. (2021) also noted in





their study that the AeroCom phase 3 models underestimate coarse AOD by 46 % when compared to 222 AERONET stations
around the globe. Finally, stations in Spain (5), the Canary Islands (7), Israel (8 and 10), Morocco (9) and Greece (13), shown
in Figure A1, indicate that the coarse AOD simulated by ARPEGE-Climat is rather close to the AERONET measurements.

The station located in the Indian Ocean (numbered 1) is predominantly influenced by sea-salt aerosols. Although AERONET
data for this station are limited (see details in Sect. 2.3), they are consistent in terms of average values with the coarse AOD
simulated by the model for this region. Nevertheless, as this coarse AOD is an order of magnitude lower than the ones from
dust, the remainder of this study will focus on the N-Africa - Arabian Peninsula region, where the coarse AOD and radiative
impact are maximum.

## 5    Radiative and climatic impacts of the LW aerosol scattering

The effects of taking into account the LW aerosol scattering on several meteorological fields, specifically high (clh, above 440
hPa) and low (cll, below 640 hPa) cloud area fractions, LW radiation at the surface (rls; net downward LW surface radiation)
and at the TOA (rlut; TOA outgoing LW radiation), and daily minimum near-surface temperature (tasmin, 2 meters, in the rest
of the paper we refer it as surface temperature), are shown in Figures 3 and 4 over our regions of interest. For information,
these results are presented at the global scale in Figure A2. To improve clarity, the results are shown for four specific months,
March, May, July and September, which correspond to the period of maximum coarse AOD (see Figure 2).

A significant increase in downwelling surface LW radiation (up to +8 W m$^{-2}$) due to the aerosol scattering in the LW
is observed in Figure 4 over much of N-Africa for the four selected months. Similarly, a significant decrease in outgoing
LW radiation at the TOA is observed, though this effect is somewhat less widespread. These changes in radiation are also
evident under clear-sky conditions, as shown in Figure A3. Our results indicate that they are even more significant in clear-sky
conditions, particularly the decrease in LW radiation at the TOA, confirming that the LW scattering of coarse aerosols has a
direct impact on radiation over N-Africa. During March and May, Figure 4 indicates significant increases in daily minimum
surface temperature over the region (up to +1.0°C), which are consistent with the rise in LW radiation at the surface. It is
interesting to note that the extent of this daily minimum surface temperature increase is much smaller than the extent of the
LW surface radiation increase. In contrast, in July and September, both Figures 4 and A3 show a drop in daily minimum and
maximum surface temperatures in the south of the N-Africa region (up to -1°C), which is significant in September. These
temperature decreases appear to be linked to a significant decrease in SW surface radiation (rss, shown in Figure A4) due to an
increase in low-troposphere cloud area fraction, as observed in Figure 3.

Further details of the annual cycle of these variables are presented in Figure 5 over three specific regions highly exposed to
dust: the Sahara (16°W - 36°E / 18°N - 30°N), the Sahel (16°W - 36°E / 10°N - 18°N) and the Arabian Peninsula (40°E - 55°E
/ 15°N - 30°N). These regions are highlighted with black frames in Figures 3 and 4. Over the Sahara and Sahel, a significant
increase in downwelling surface LW radiation (+5 W m$^{-2}$ on average) is observed from February to October, as shown in
Figure 5. Changes in radiation, particularly outgoing LW radiation at the TOA, due to clouds (the difference between all-sky
and clear-sky fluxes) appear to be mostly correlated with changes in high troposphere cloud area fraction. In fact, the most




important changes in radiation associated with clouds occur in July and August over the Sahara, in April over the Sahel and in April and August over the Arabian Peninsula, coinciding with the peak increases in high troposphere cloud area fraction over these regions. Figure 5 also highlights an increase in daily minimum surface temperature during April, May, June and July over

the Sahara (+0.5°C on average), and during March (+1.0°C) and April (+0.5°C) over the Sahel. These temperature increases are directly linked to the changes in surface LW radiation over these regions. In contrast, during July, August and September, Figure 5 shows a significant decrease of the daily minimum (-0.5°C) and maximum (-0.8°C) surface temperatures over the Sahel. This figure also shows that this temperature drop is the result of a significant drop in downwelling SW surface radiation due to a significant increase in low troposphere cloud area fraction resulting from stronger atmospheric stratification (not shown

here). Additionally, including aerosol scattering in the LW spectral range leads to a significant increase in precipitation over the Sahel during September (+0.6 mm day$^{-1}$), as shown in Figure A5 (note that convective precipitations are identical to total precipitations in this region). This increase is likely due to wetter lower atmospheric layers which are more likely to precipitate (not shown here). Lastly, over the Arabian Peninsula region, Figure 5 shows significant increases in daily minimum surface temperature during March (+1°C), and also during May, June and July (+0.5°C), which can be attributed to the increased LW

radiation at the surface from aerosol scattering. It is noteworthy that the greatest direct impact is on daily minimum surface temperatures (which occurs during the night), which are more influenced by changes in LW radiation than on daily maximum surface temperatures, which are more influenced by SW radiation.

Annual averages of the LW radiation at the TOA and at the surface in all-sky and clear-sky conditions, over the three regions studied, are summarized in Table 1. Neglecting LW aerosol scattering (NOLWAS simulation) results in a net LW radiation

underestimation between -3.1 and -4.3 W m$^{-2}$ (i.e. between 2.6 and 3.5% of the total) at the surface and between -3.9 and -4.4 W m$^{-2}$ (i.e. between 1.4 and 1.6% of the total) at the TOA in all-sky conditions. In clear-sky conditions, LW scattering of aerosols has less impact on radiation. Indeed, in clear-sky conditions, neglecting LW aerosol scattering results in a net LW radiation underestimation between -2.7 and -4.0 W m$^{-2}$ (i.e. between 2.2 and 3.2% of the total) at the surface and between -2.9 and -3.7 W m$^{-2}$ (i.e. between 1.0 and 1.3% of the total) at the TOA. These results are consistent with those of the findings

of Dufresne et al. (2002) for the tropical and dry tropical atmospheric profiles, as summarized in Table 1. Specifically, they showed that neglecting LW aerosol scattering could lead to a radiative forcing error in the LW range comprised between - 3.5 and - 5.3 W m$^{-2}$ at the surface and between - 3.5 and - 4.9 W m$^{-2}$ at the TOA, which is in close agreement with our results. However, it is important to note that the configuration of our simulations does not allow for a direct calculation of the aerosol LW radiative forcing. Consequently, the radiative flux differences presented here account for both changes in aerosol

radiative forcing and changes in weather. Dufresne et al. (2002) also showed that LW scattering can have varying impacts on radiation depending on the thickness and altitude of the aerosol layer, which may explain the greater impact at the surface (in percentages) than at the TOA in our study.

On a larger scale (Figure A2 and Table 1), our study shows a global annual mean contribution of aerosol LW scattering of +0.4 W m$^{-2}$ at the surface and of -0.4 W m$^{-2}$ at the TOA (i.e., net LW radiation at TOA of +0.4 W m$^{-2}$) in all-sky conditions.

We note that the areas statistically significant mostly correspond to those analyzed above, i.e., the Sahara, Sahel and Arabian Peninsula. In comparison, studies by Di Biagio et al. (2020) and Hogan and Bozzo (2018) suggest a weaker impact from



aerosol LW scattering. Indeed, Hogan and Bozzo (2018) concluded that turning on aerosol LW scattering has an impact on the global mean net LW irradiances only up to +0.1 W m$^{-2}$ at the surface and the TOA. In clear-sky conditions (without cloud contribution but accounting for other weather changes), the global annual mean contribution of aerosol LW scattering (+0.3 W m$^{-2}$ at the surface and TOA) is closer to the values reported by Hogan and Bozzo (2018), even though it remains three times higher. For their part, Di Biagio et al. (2020) highlighted a global annual mean all-sky DRE of desert dust of +0.22 W m$^{-2}$ at TOA in the LW range, with a LW scattering contribution of 51%. However, Di Biagio et al. (2020) calculated a lower estimate of the LW scattering by coarse dust as they applied a LW DRE correction that accounted only for the LW scattering of dust with a diameter smaller than 10 μm. Additionally, the use of different aerosol optical properties may also partially explain the discrepancies between our results and their study.

## 6 Conclusions

Aerosol scattering in the LW spectrum is still very often neglected in climate models, both at regional and global scales, despite several studies highlighting the significance of this process for large particles such as desert dust. To date, when LW aerosols scattering is not completely neglected, it is typically treated in a simplistic manner by applying constant correction factors to "artificially" increase the AOD or the retrieved TOA direct radiative effect.

In contrast to previous approaches, we have been able to analyze impacts of the LW aerosol scattering physically modeled in a global climate atmospheric model. For this, we conducted simulations with the CNRM ARPEGE-Climat model using a specific unique feature of its radiation scheme ecRad (Hogan and Bozzo, 2018) which allows for turning on or off the 3D scattering of aerosols in the LW. We analyzed climatological results from 30 year-long CMIP6 amip-type simulations (1985-2014) over the globe, with a focus on three regions characterized with the highest coarse aerosol AOD.

We revised the optical properties of coarse aerosols, especially those of dust, for the sixteen LW spectral bands of ecRad. Our updated dust refractive indices are derived from multiple studies: Di Biagio et al. (2017) for wavelengths ranging from 3 to 15 μm, and Woodward (2001) for longer wavelengths, 15 to 40 μm.

The ARPEGE-Climat coarse AOD climatology over North Africa and the Arabian Peninsula successfully captures the spatio-temporal variations of the AERONET data, with an average correlation coefficients of 0.86. Coarse AOD biases range from -0.11 to +0.10, depending on the station's location. In the northern part of these regions, the model generally overestimates coarse AOD of about 30 to 40 % (+0.10) throughout the year while over the southern part of these regions it underestimates coarse AOD by approximately 40 to 50% (-0.10).

Over the three regions examined in this study (the Sahara, Sahel and Arabian Peninsula), accounting for aerosol scattering in the LW spectral range leads to a significant increase in surface LW radiation (+5 W m$^{-2}$ on average) from March to September. This change induces notable temperature increases, particularly in daily minimum surface temperature (ranging from +0.5 and +1 °C depending on the region and time period), which are directly linked to the rise in LW radiation at the surface. Conversely, in certain months, both daily minimum and maximum surface temperatures drop significantly (up to -0.8 °C). These temperature decreases are associated with a significant reduction in SW surface radiation, which is due to a

significant increase in low troposphere cloud area fraction. This increase in low clouds is the result of stronger stratification in the lower troposphere. Finally, a significant increase in precipitation, linked to wetter lower atmospheric layers, is also observed over the Sahel in September.

The results presented in this study highlight the importance of incorporating aerosol scattering in the LW spectrum in climate models. However, improving the coarse AOD modeling, by enhancing dynamical emissions reducing potential wind or rain

biases in the model, could further improve the model in terms of both radiative and climatic impacts. Additionally, while several studies have already been published, further research is needed to improve the knowledge about the refractive index of large particles such as desert dust or sea-salt, and more broadly their optical properties. Using an interactive aerosol scheme would also provide a better representation of the spatio-temporal variability of aerosols in the model and allow for the study of specific events such as intense dust episode or heat waves. Finally, fully coupled atmosphere-ocean climate model simulations

would be relevant for enabling LW aerosol scattering to influence air-sea interactions.

*Data availability.* This study relies entirely on publicly available data, which is available at: https://doi.org/10.5281/zenodo.14186858 (Drugé et al., 2024). AERONET data are available at https://aeronet.gsfc.nasa.gov/ (last access: 30 October 2024).

*Author contributions.* All authors designed the experiment methodology and TD carried them out. TD wrote the paper with contributions from all co-authors.

*Competing interests.* The authors declare that they have no conflict of interest.

*Acknowledgements.* This work has received funding from the European Union's Horizon 2020 research and innovation programme under Grant Agreement N° 101003536 (ESM2025–Earth System Models for the Future) as well as from the European Union's Horizon innovation programme under grant agreement N° 101081193 (OptimESM-Optimal High Resolution Earth System Models for Exploring Future Climate

Change). We thank the principal investigators of the AERONET network and his staff for establishing and maintaining the different sites used in this investigation. We also acknowledge the support of the entire team in charge of the CNRM climate models and in particular Romain Roehrig and Yves Bouteloup for their contribution. Supercomputing time was provided by the Météo-France/DSI supercomputing center.



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





|  |  | Long-wave radiation at the TOA (rlut) | | | | Long-wave radiation at the surface (rls) | | | |
|---|---|---|---|---|---|---|---|---|---|
|  |  | LWAS (W m$^{-2}$) | NOLWAS (W m$^{-2}$) | Error (W m$^{-2}$) | Relative error (%) | LWAS (W m$^{-2}$) | NOLWAS (W m$^{-2}$) | Error (W m$^{-2}$) | Relative error (%) |
| All-sky | Sahara | 271.9 | 276.3 | 4.4 | 1.6 | -111.7 | -115.8 | -4.1 | 3.5 |
|  | Sahel | 260.8 | 264.9 | 4.1 | 1.5 | -95.2 | -99.5 | -4.3 | 4.3 |
|  | Arabian Peninsula | 276.6 | 280.5 | 3.9 | 1.4 | -114.2 | -117.3 | -3.1 | 2.6 |
|  | Global | 201.1 | 201.5 | 0.4 | 0.2 | -57.5 | -57.9 | -0.4 | 0.7 |
| Clear-sky | Sahara | 289.7 | 293.4 | 3.7 | 1.3 | -117.7 | -121.5 | -3.8 | 3.2 |
|  | Sahel | 290.0 | 293.1 | 3.1 | 1.1 | -103.3 | -107.3 | -4.0 | 3.9 |
|  | Arabian Peninsula | 291.7 | 294.6 | 2.9 | 1.0 | -120.8 | -123.6 | -2.7 | 2.2 |
|  | Global | 218.9 | 219.2 | 0.3 | 0.1 | -81.6 | -81.9 | -0.3 | 0.4 |

| Dufresne et al. 2002 | Aerosol long-wave radiative forcing at the TOA | | | | Aerosol long-wave radiative forcing at the surface | | | |
|---|---|---|---|---|---|---|---|---|
| Tropical | X | X | -3.5 | X | X | X | -3.5 | X |
| Dry Tropical | X | X | -4.9 | X | X | X | -5.3 | X |

**Table 1.** Annual mean (1985-2014) of the LW radiation at the TOA (rlut) and at the surface (rls) in all-sky and clear-sky conditions over the three regions considered in this study, Sahara (16°W-36°E / 18°N-30°N), Sahel (16°W-36°E / 10°N-18°N) and Arabian Peninsula (40°E-55°E / 15°N-30°N), compared with the error not considering aerosol LW radiative forcing for tropical and dry tropical atmospheric profiles in the study of Dufresne et al. (2002).



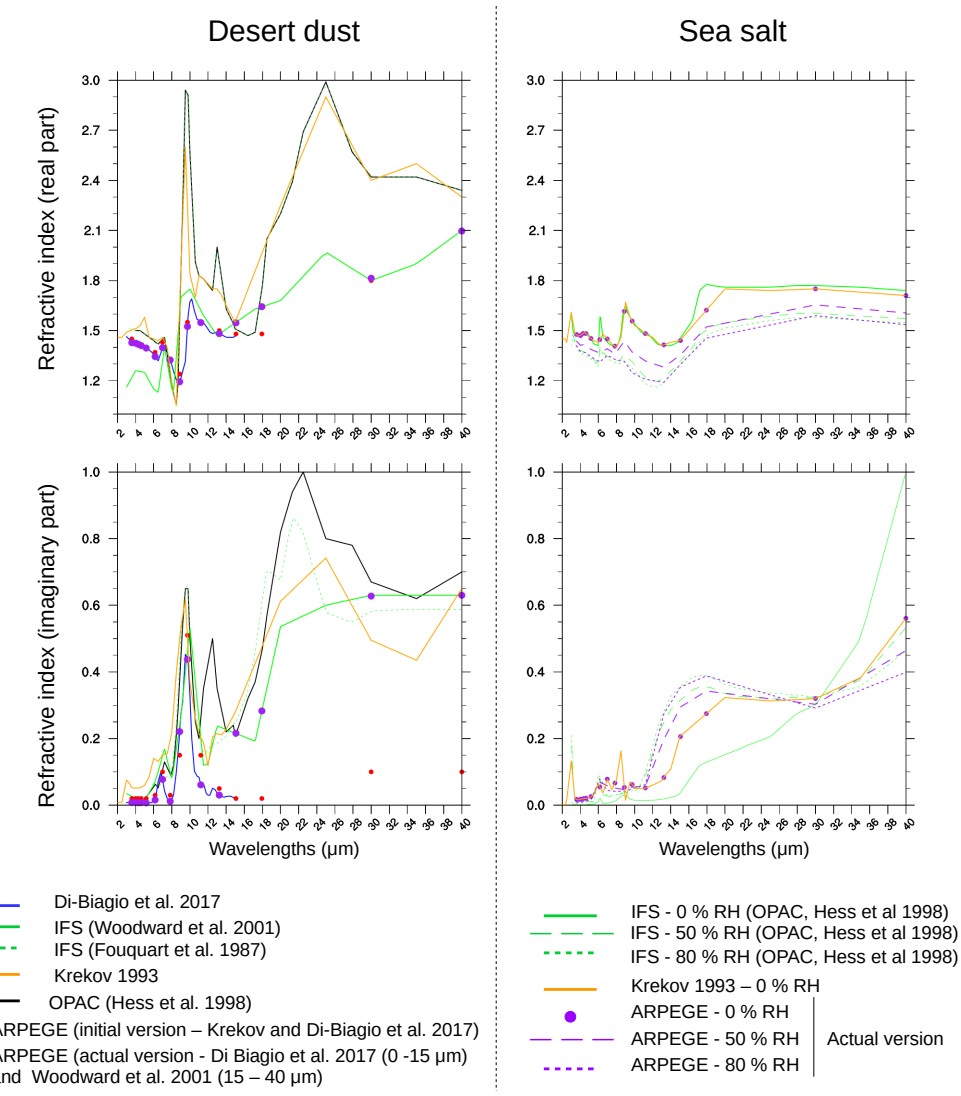

**Figure 1.** Real (first line) and imaginary (second line) refractive indices of desert dust (left column) and sea-salt (right column) particles. The optical properties used in this study were calculated from the refractive indices shown in purple.



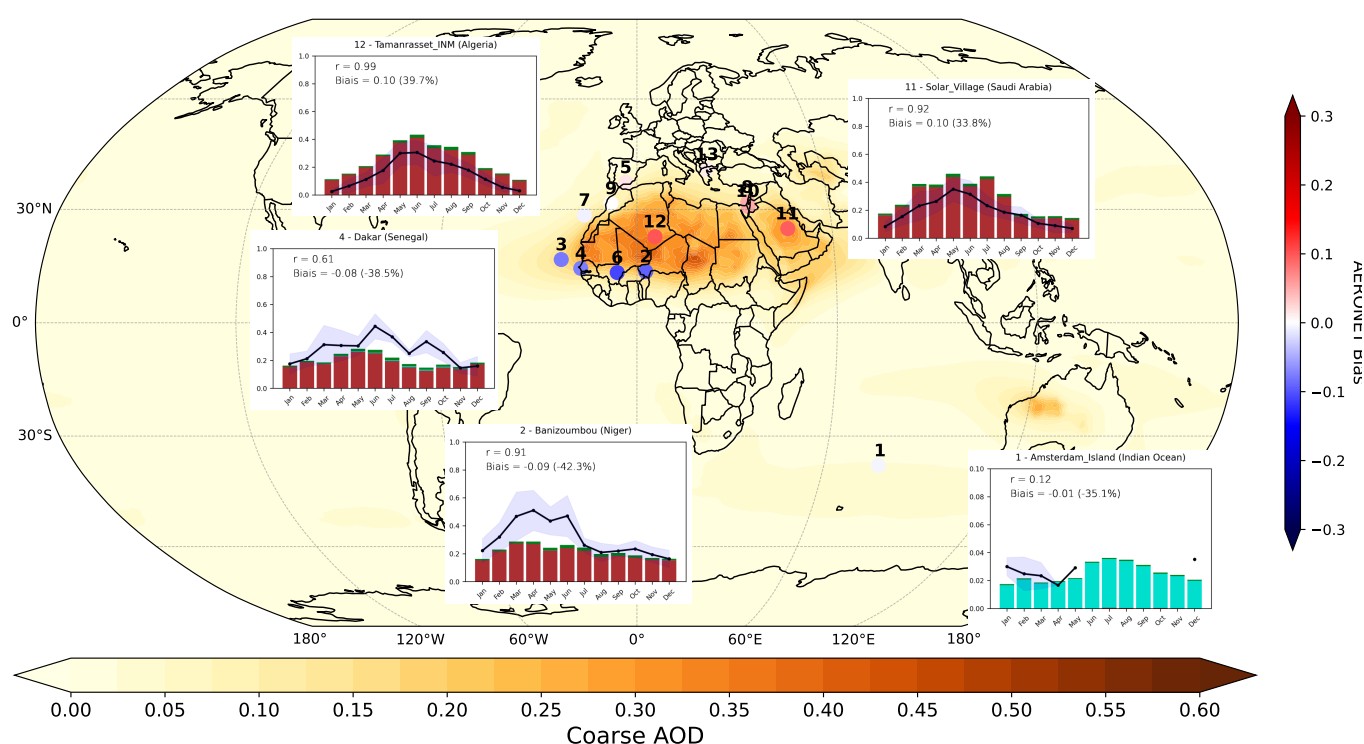

**Figure 2.** Coarse AOD (550 nm, annual mean over 1985-2014) simulated by the ARPEGE-Climat model. AERONET stations are represented by circles (see Table A1 for details on these stations). The color of the circles corresponds to the coarse AOD bias between the model and the station. Annual cycles are shown for five AERONET stations, compared with AERONET measurements (black, with standard deviation in light blue). Coarse desert dust is shown in brown, coarse sea-salt in cyan and coarse nitrate in green.



**Figure 3.** Mean differences (1985-2014) between the LWAS and NOLWAS simulations (LWAS minus NOLWAS) in coarse AOD (550 nm, left column), and cloud area fraction (%), at high (above 440 hPa, middle column), and low (below 640hPa, right column) altitudes (clh and cll, respectively), for the months of March (1st line), May (2nd line), July (3rd line) and September (4th line). Hatching indicates regions with a significant effect at the 0.05 level (Student's t-test).







**Figure 4.** Same as in Figure 3 for rls (net downward LW surface radiation, rls = rlds - rlus, W m$^{-2}$), rlut (TOA outgoing LW radiation, W m$^{-2}$) and tasmin (minimum surface temperature, K).



**Figure 5.** Annual (1985-2014) cycles of changes in coarse AOD (550 nm), rls (net downward LW surface radiation, rls = rlds - rlus, W m$^{-2}$), rlut (TOA outgoing LW radiation, W m$^{-2}$), rss (net downward SW surface radiation, rss = rsds - rsus, W m$^{-2}$), clh and cll (high, above 440 hPa, and low, below 640hPa, troposphere cloud area fraction, %) changes and tasmax and tasmin changes (maximum and minimum surface temperature, K) between the LWAS and NOLWAS simulations (LWAS minus NOLWAS) over three regions: Sahara, Sahel and Arabian Peninsula. No significant changes in confidence intervals indicated in light color (Student's t-test, 0.05 level).





**Appendix A**

| Station | Location | Altitude (m) | Number of months 3 years | Total years available |
|---|---|---|---|---|
| 1 - Amsterdam_Island (Indian Ocean) | 37.8S, 77.6E | 49 | 6 (no criteria) | 8 |
| 2 - Banizoumbou (Niger) | 13.5N, 2.7E | 274 | 12 | 20 |
| 3 - Capo_Verde (Cape Verde) | 16.7N, 22.9W | 60 | 9 | 21 |
| 4 - Dakar (Senegal) | 14.4N, 17.0W | 21 | 12 | 19 |
| 5 - Granada (Spain) | 37.2N, 3.6W | 680 | 12 | 16 |
| 6 - IER_Cinzana (Mali) | 13.3N, 5.9W | 285 | 12 | 16 |
| 7 - La_Laguna (Canary Islands, Spain) | 28.5N, 16.3W | 568 | 11 | 14 |
| 8 - Nes_Ziona (Israël) | 31.9N, 34.8E | 40 | 12 | 13 |
| 9 - Saada (Morocco) | 31.6N, 8.2W | 420 | 12 | 17 |
| 10 - SEDE_BOKER (Israël) | 30.9N, 34.8E | 480 | 12 | 21 |
| 11 - Solar_Village (Saudi Arabia) | 24.9N, 46.4E | 764 | 12 | 16 |
| 12 - Tamanrasset_INM (Algeria) | 22.8N, 5.5E | 1377 | 12 | 15 |
| 13 - Thessaloniki (Greece) | 40.6N, 23.0E | 60 | 12 | 15 |

**Table A1.** Characteristics of the AERONET stations used in this study: station name, location, altitude, number of months with data available over at least 3 years during the observation period (2000-2020), and total years available over the observation period (2000-2020).





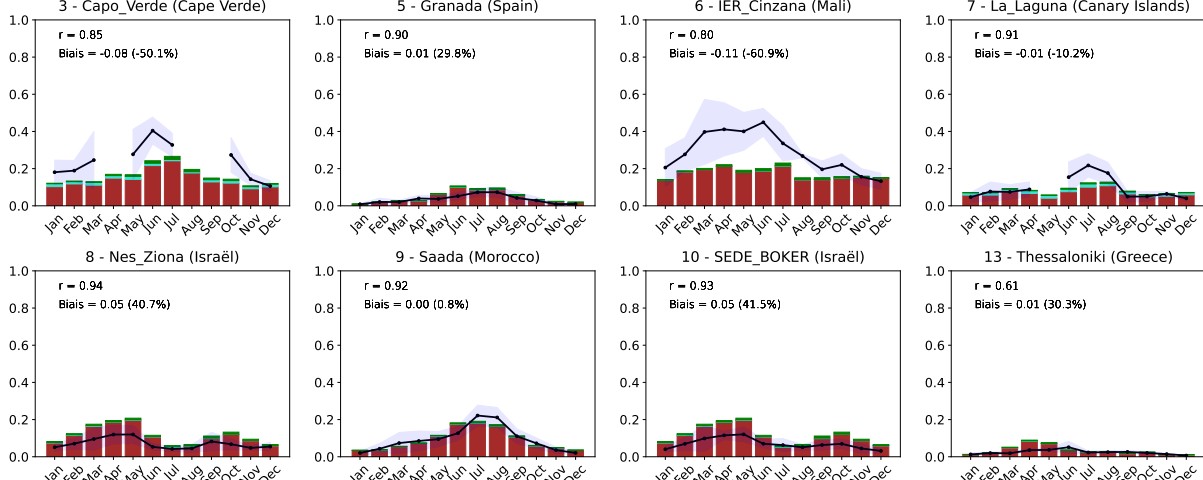

**Figure A1.** Annual cycles of coarse AOD simulated by the ARPEGE-Climat model (desert dust in brown, sea-salt in cyan and nitrate in green), compared with AERONET measurements (black, with standard deviation in light blue). The locations of the stations are shown in Figure 2 (see Table A1 for details on these AERONET stations).





**Figure A2.** Changes (1985-2014), at the global scale, between the LWAS and NOLWAS simulations (LWAS minus NOLWAS) in rls (net downward LW surface radiation, rls = rlds - rlus, W m$^{-2}$), rlut (TOA outgoing LW radiation, W m$^{-2}$) and tasmin (minimum surface temperature, K) for the months of March (1st line), May (2nd line), July (3rd line) and September (4th line). Hatching indicates regions with a significant effect at the 0.05 level (Student's t-test).



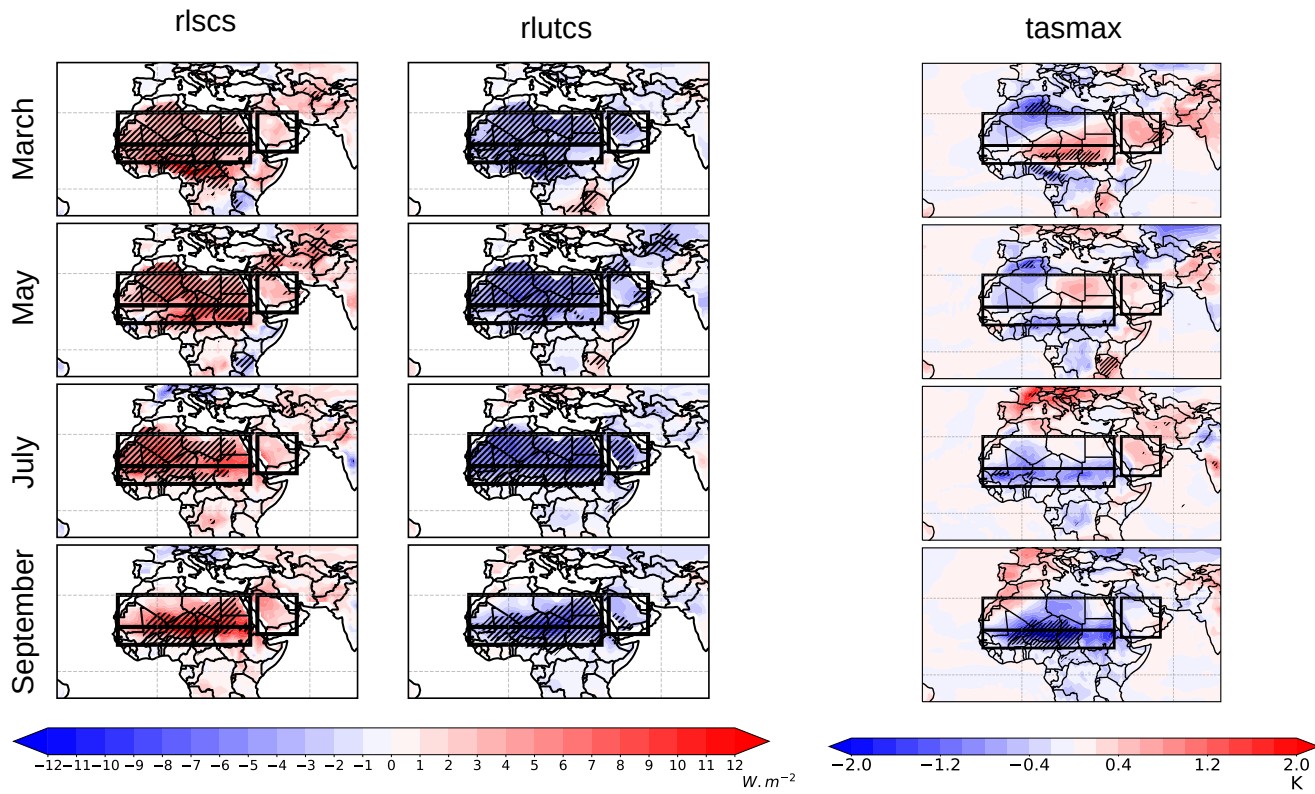

**Figure A3.** Same as Figure A3 for rlscs (net downward LW surface radiation in clear-sky conditions, W m$^{-2}$), rlutcs (TOA outgoing LW radiation in clear-sky conditions, W m$^{-2}$) and tasmax (maximum surface temperature, K) for the months of March (1st line), May (2nd line), July (3rd line) and September (4th line). Hatching indicates regions with a significant effect at the 0.05 level (Student's t-test).





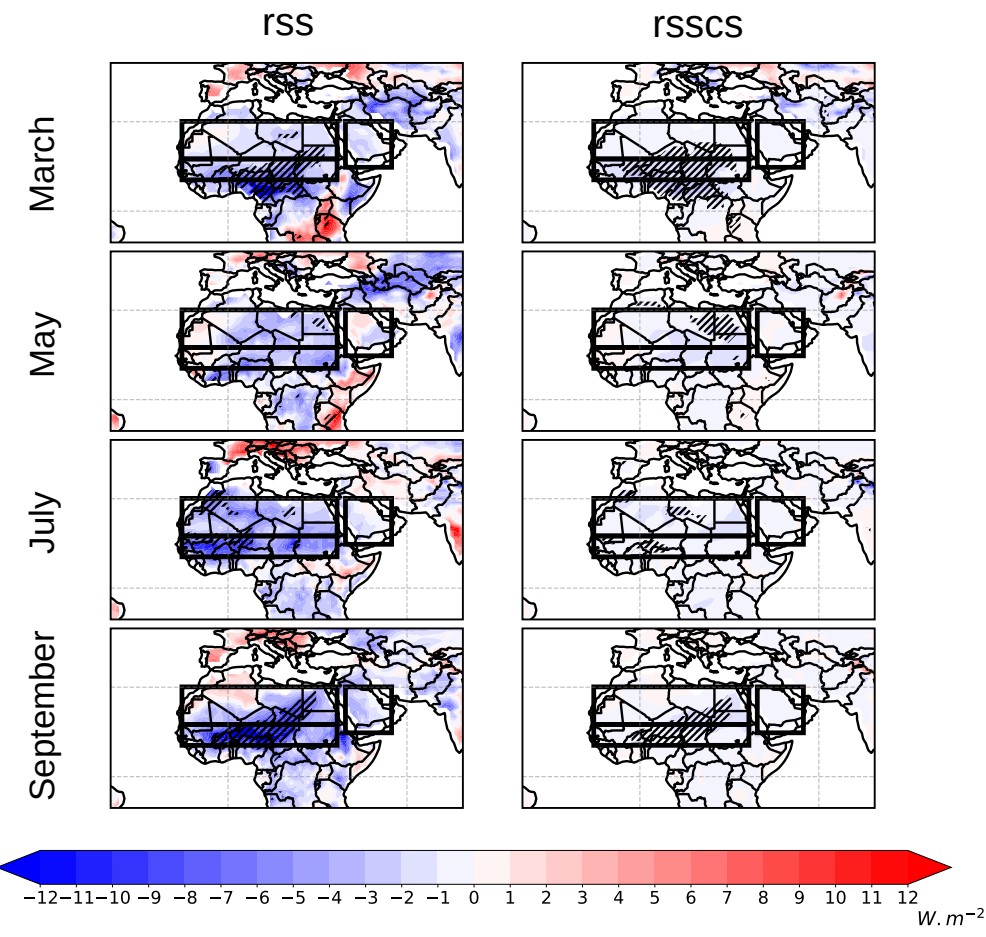

**Figure A4.** Same as Figure A3 but for rss (net downward SW surface radiation, rss = rsds - rsus, W m$^{-2}$) and rsscs (net downward SW surface radiation in clear-sky conditions, W m$^{-2}$).





**Figure A5.** Same as Figure A3 but for pr (precipitations, mm day$^{-1}$).