# Peer review of "Radiative and climate effects of aerosol scattering in long-wave radiation based on global climate modeling"

_EGUsphere, 2024_

## Referee Comment (RC1)

**Review of the paper "Radiative and climate effects of aerosol scattering in long-wave radiation based on global climate modeling" by Drugé et al.**

The manuscript explores the impact of neglecting the representation of aerosol scattering in the longwave spectrum on the simulations of a Global Climate model. The radiative impact of longwave aerosol scattering is analysed using a 30-year integration of the ARPEGE-Climat atmospheric global climate model with prescribed boundary conditions. The authors analyse a set of model variables and their change between the simulation that includes longwave scattering and the simulation neglecting it.

The topic is of certain interest to improve the quantification of the uncertainty in aerosol radiative forcing on the Earth radiative budget and of the role of assumptions in the radiative transfer modelling. The paper is well organised and the methodology sufficiently clear.

My main issue with the current version of the manuscript is a lack of in-depth analysis of some of the main impacts observed when aerosol longwave scattering is enabled in the radiative computations. In particular, the change in low and high cloud fraction displayed by the model is puzzling because no mechanism behind it is discussed. Previous studies (e.g. Dufresne et al. 2002) suggest that the impact of longwave aerosol scattering on the heating rate profiles is relatively small: is this the case also in this study? Modification in the lower atmospheric stability is mentioned as one reason to explain the increase in low level cloud fraction. It would be interesting to see model data to support this hypothesis. Are surface temperature changes such as the ones reported here, enough to sustain the observed modification in the cloud fraction? Or is there any other feedback at play?

I think that to support the conclusion that climate simulations should explicitly include longwave aerosol scattering, a more complete description of its impacts on the model fields would be helpful.

The language throughout the text is generally clear, but I suggest a double check to improve the text in places (few suggestions in the specific comments below).

**Specific comments:**

**Abstract, line 1**: "The few studies that considered aerosol scattering in the long-wave (LW) typically relied on artificially increasing it." Please explain or rephrase, it is not clear what it is meant by "artificially increasing it"

**Abstract, line 8**: "in line with the maximum coarse AOD" please clarify, e.g. "correlated with the largest AOD from coarse particles."

**Abstract, line 10**: "However, during certain months and regions" -> "However, during certain months and in certain regions"

**Introduction, page 1, line 22**: please indicate the reference for AR5. Also, AR6 should be explained above, where there is the first reference for the latest IPCC Assessment Report.

**Introduction, page 3, line 64**: Values in Dufresne et al. (2002) are for specific profiles and not global means though, right?

**Section 2.2:** In the discussion of the results, the impact of the longwave aerosol scattering is shown for all-sky and clear sky conditions. Perhaps it could be mentioned here how the two contributions are calculated (i.e. selecting clear sky areas or with separated clear sky computations?)

**Section 3, line 192:** Differences in the RI for wavelengths above 20 microns is likely of minor importance: do spectral regions outside the IR window contribute significantly to the results shown in this study? Generally, only for extremely dry atmospheres the radiative effect of longwave scattering from aerosols is significant for spectral regions above 20 microns.

**Section 3, line 196:** Since the sea salt results are not discussed, perhaps this can be removed/shortened?

**Section 5, line 251:** Please clarify what is it meant "this daily minimum surface temperature increase is much smaller than the extent of the LW surface radiation increase." Does this refer to the area showing changes respectively in LW fluxes and surface temperature or the magnitude?

**Section 5, line 254 and 269:** These results could be interesting and deserve more in-depth analysis in my opinion. Is there an interaction with the strength/position of the ITCZ or/and the West African Monsoon region? Also, this is the region where there is a significant negative bias in AOD compared to AERONET, does this have an impact on the results?

**Section 5, line 282:** "In clear-sky conditions, LW scattering of aerosols has less impact on radiation." Doesn't this contradict what said at line 247 "Our results indicate that they are even more significant in clear-sky conditions?" Please clarify.

**Section 5, line 290:** This makes the simulation results interesting but somewhat difficult to interpret. As far as possible within the current simulation, physical interactions between model variables should be analysed.

**Conclusions, line 313:** "for turning on or off the 3D scattering of aerosols in the LW." What is it meant by 3D scattering?

**Conclusions, line 313:** This information is mostly auxiliary to the current study, and I would not mention it in the main conclusions which should be focussed on the mean results of the impact of the explicit treatment of longwave scattering in aerosol radiative effects.

**Conclusions, line 334-338:** An estimate of the different weight of these contributions to the general longwave aerosol radiative forcing would be interesting, to put in context the size of the effect analysed here. It could be added to the main discussion.

**Figures**

**Figure 3:** From the caption it sounds like the first column shows AOD differences, which had me confused. Please rephrase.

**Figure 4:** the symbols rlds and rlus have never been defined.

**Figure 5:** The colours in the second panel are not very clear and it is difficult to discern the various lines. It is also not very clear what it is meant by "No significant changes in confidence intervals indicated in light color". Is the light colour shading indicating the confidence interval or the significance of the differences between the two runs? How is this computed? Please clarify here or in the mean methods section.

**Figure A3:** it should be "same as Figure 4"

---

## Author Comment (AC1)

We would like to thank the anonymous referee for his comments mentioning different points listed below. The reviewer's comments are in black, and the answers are in red. New information and explanations in the new version of the article are italicized.

**Anonymous Referee 1**

The manuscript explores the impact of neglecting the representation of aerosol scattering in the longwave spectrum on the simulations of a Global Climate model. The radiative impact of longwave aerosol scattering is analysed using a 30-year integration of the ARPEGE-Climat atmospheric global climate model with prescribed boundary conditions. The authors analyse a set of model variables and their change between the simulation that includes longwave scattering and the simulation neglecting it.

The topic is of certain interest to improve the quantification of the uncertainty in aerosol radiative forcing on the Earth radiative budget and of the role of assumptions in the radiative transfer modelling. The paper is well organised and the methodology sufficiently clear.

My main issue with the current version of the manuscript is a lack of in-depth analysis of some of the main impacts observed when aerosol longwave scattering is enabled in the radiative computations. In particular, the change in low and high cloud fraction displayed by the model is puzzling because no mechanism behind it is discussed. Previous studies (e.g. Dufresne et al. 2002) suggest that the impact of longwave aerosol scattering on the heating rate profiles is relatively small: is this the case also in this study? Modification in the lower atmospheric stability is mentioned as one reason to explain the increase in low level cloud fraction. It would be interesting to see model data to support this hypothesis. Are surface temperature changes such as the ones reported here, enough to sustain the observed modification in the cloud fraction? Or is there any other feedback at play?

Thanks for this suggestion, we acknowledge the first version submitted did not provide in-depth analysis of the impacts on climate variables, notably to explain the changes noted in the Sahel in September. Therefore, the ARPEGE simulations have been relaunched to get additional diagnostics and thus providing new answers. The vertical velocity (wap), temperature and specific humidity vertical profiles are presented below in Figure 1. This figure shows that taking into account the LW scattering of aerosols leads to a significant reduction in temperature below 700 hPa, which in turn tends to stabilize these lower atmospheric layers. Indeed, negative values of wap correspond to convection, and as these values are less negative with the LWAS simulation below 700 hPa, this reflects a decrease in convection when the LW aerosol scattering process is activated. This drop in low-level convection, combined with a significant rise in humidity, has resulted in a stabilization of the lowest atmospheric layers and an increase in low clouds over the Sahel in September. Conversely, above 700 hPa, we can observe a rise in temperature, induced by LW aerosol scattering, which leads to a significant increase in convection. This increase in convection, coupled as before with a rise in humidity, favors high clouds and convective rain over the Sahel in September.

The new version includes these explanations as follows:

Page 9: *"Figure A6, which presents the vertical velocity (wap), temperature and specific humidity vertical profiles over the Sahel during September, shows that the significant reduction in temperature below 700 hPa over this region reduces convection in the lowest atmospheric layers (wap values are less negative in the LWAS simulation below 700 hPa). This drop in low-level convection, combined with a significant rise in humidity, has resulted in a stabilization of the lowest atmospheric layers and an increase in low clouds and convective rain over the Sahel in September. Conversely, above 700 hPa, Figure A6 highlights a significant temperature rise, which leads to a significant increase in convection. This stronger convection, coupled with a humidity augmentation, also favors high clouds and convective rain over the Sahel in September."*

The impact of LW aerosol scattering on the LW heating rate (named tntrl) profiles has also been studied and summarized in Figure 2 below. This figure clearly shows a small impact of the LW aerosol scattering on this heating rate, and this result is consistent with the study of Dufresne et al. 2002. The only significant impact is between the surface and 700 hPa (consistent with the vertical coarse aerosol concentration profile, see also Figure 2) during July, which is the month with the highest AOD (0.55). On the other hand, no significant change is visible in September (month studied in detail in the article). Compared to the study of Dufresne et al. 2002, the tntrl profiles show lower values (in absolute terms) because Dufresne et al. 2002 provide tntrl for an AOD of 1, whereas our tntrl correspond to an AOD of 0.55 (July) and 0.47 (September). The use of different asymmetry parameters between these two studies may also partly explain these differences in tntrl.

The new version includes these explanations as follows:
Page 9: *"The impact of the aerosol scattering in the LW spectrum on the LW heating rate (named tntrl) has also been studied*
*over the Sahara region (see Figure 2), and is found to be relatively weak, which is consistent with the study of Dufresne et al. 2002. The only significant impact is between the surface and 700 hPa, consistent with the vertical coarse aerosols concentration profile, in July, which is the month with the highest AOD (0.55) over the Sahara region. On the other hand, no significant change is visible in September."*

These two figures have also been added in the appendix to the article.

I think that to support the conclusion that climate simulations should explicitly include longwave aerosol scattering, a more complete description of its impacts on the model fields would be helpful.

Radiative and climatic impacts of aerosol scattering in the LW spectrum are now better understood and described in the article, thanks to new figures showing vertical profiles of LW heating rate, vertical velocity, temperature and humidity. In addition to the elements in response to the first comment above, a sentence has been added in the conclusion to mention the impact of convection changes: *"This increase in low clouds is the result of stronger stratification in the lower troposphere, which is a consequence of weaker convection at these altitudes."*

The language throughout the text is generally clear, but I suggest a double check to improve the text in places (few suggestions in the specific comments below).

The text has been rechecked and enriched with suggestions from the two reviewers.

Specific comments:

Abstract, line 1: "The few studies that considered aerosol scattering in the long-wave (LW) typically relied on artificially increasing it." Please explain or rephrase, it is not clear what it is meant by "artificially increasing it"

This sentence has been clarified: *"The few studies that considered aerosol scattering in the long-wave (LW) typically relied on using simple corrective factors instead of including it in the radiative code."*

Abstract, line 8: "in line with the maximum coarse AOD" please clarify, e.g. "correlated with the largest AOD from coarse
particles."

This sentence has been clarified: *"correlated with the largest coarse AOD."*

Abstract, line 10: "However, during certain months and regions" -> "However, during certain months and in certain regions"
Done.

Introduction, page 1, line 22: please indicate the reference for AR5. Also, AR6 should be explained above, where there is the first reference for the latest IPCC Assessment Report.
Done.

Introduction, page 3, line 64: Values in Dufresne et al. (2002) are for specific profiles and not global means though, right?

This is right, and has been clarified in the text. *"which is far from the 3 to 5 W m-2 cited in Dufresne et al. (2002) for standard vertical profiles"*

Section 2.2: In the discussion of the results, the impact of the longwave aerosol scattering is shown for all-sky and clear sky conditions. Perhaps it could be mentioned here how the two contributions are calculated (i.e. selecting clear sky areas or with
separated clear sky computations?)

New information on clear-sky and all-sky diagnostics has been added to section 2.2: *"The various radiative diagnostics provided in this study have been computed in all-sky conditions and in clear-sky ones, as classically done. In clear-sky conditions, only clouds are removed, surface temperatures and water vapor remaining unchanged."*

Section 3, line 192: Differences in the RI for wavelengths above 20 microns is likely of minor importance: do spectral regions outside the IR window contribute significantly to the results shown in this study? Generally, only for extremely dry atmospheres the radiative effect of longwave scattering from aerosols is significant for spectral regions above 20 microns.

Indeed, the LW aerosol radiative forcing is maximum for wavelengths below 20 microns. However, Sicard et al. (2014) also indicated that large particles have a non-negligible effect in the 17 to 22 $\mu$m range at the TOA. It was therefore important to update our desert dust IR values over the entire spectrum of long wavelengths, from 3 to 40 $\mu$m.

Section 3, line 196: Since the sea salt results are not discussed, perhaps this can be removed/shortened?

This part about sea-salt RI has been shortened.

Section 5, line 251: Please clarify what is it meant "this daily minimum surface temperature increase is much smaller than the extent of the LW surface radiation increase." Does this refer to the area showing changes respectively in LW fluxes and surface
temperature or the magnitude?

This sentence has been clarified and replaced by: *"It is interesting to note that the area covered by this daily minimum surface temperature increase is much smaller than the area covered by the LW surface radiation increase."*

Section 5, line 254 and 269: These results could be interesting and deserve more in-depth analysis in my opinion. Is there an interaction with the strength/position of the ITCZ or/and the West African Monsoon region? Also, this is the region where there is a significant negative bias in AOD compared to AERONET, does this have an impact on the results?

Indeed, aerosols influence the dynamics of the African monsoon, particularly the West African Monsoon, through various
mechanisms such as radiative forcing, atmospheric circulation or cloud microphysics (Roehrig et al. 2013, Solmon et al. 2021). For example, biomass burning aerosols from Southern Africa have been shown to affect the West African Monsoon by inducing regional scale and inter-hemispheric dynamical feedbacks (Solmon et al. 2021). These feedbacks can alter the strength and position of the monsoon system. Moreover, Solmon et al. 2008 have shown that Lw scattering by aerosols leads to heating of the atmospheric layer where aerosols are concentrated, which can stabilize the atmosphere, suppress convection
and reduce monsoon rainfall. A further study on the LW aerosol scattering impacts on the African monsoon would therefore be interesting. The AOD bias present over this region may also have an impact on the results presented here and reducing this bias would therefore allow for a better estimate of the impact of LW aerosol scattering.

Section 5, line 282: "In clear-sky conditions, LW scattering of aerosols has less impact on radiation." Doesn't this contradict
what said at line 247 "Our results indicate that they are even more significant in clear-sky conditions?" Please clarify.

*Indeed, these two sentences may seem contradictory. Line 247, the analysis is based on the months of March, May, July and September, whereas line 282 we refer to the annual average. These sentences have been clarified: "Our results indicate that they are even more significant in clear-sky conditions over these four months" and "In clear-sky conditions, and as an annual average, LW scattering of aerosols has less impact on radiation."*

Section 5, line 290: This makes the simulation results interesting but somewhat difficult to interpret. As far as possible within the current simulation, physical interactions between model variables should be analysed.

*As explained above, new simulations have been carried out to provide new diagnostics (LW heating rates, vertical velocity, temperature and specific humidity vertical profiles) and have enabled us to go further in our analysis (see our response to the general comment).*

Conclusions, line 313: "for turning on or off the 3D scattering of aerosols in the LW." What is it meant by 3D scattering?

*This sentence has been deleted in agreement with the following comment.*

Conclusions, line 313: This information is mostly auxiliary to the current study, and I would not mention it in the main conclusions which should be focussed on the mean results of the impact of the explicit treatment of longwave scattering in aerosol radiative effects.

*This part has been simplified and shortened.*

Conclusions, line 334-338: An estimate of the different weight of these contributions to the general longwave aerosol radiative forcing would be interesting, to put in context the size of the effect analysed here. It could be added to the main discussion.

*We agree it might be interesting to know which of these suggestions is the most promising, but it is difficult to estimate. Furthermore, these lines were more intended to suggest different ways of improving our climate models. Probably the results would depend on each model.*

Figures

Figure 3: From the caption it sounds like the first column shows AOD differences, which had me confused. Please rephrase.

*The legend of the figure 3 has been rephrased. "Coarse AOD (550 nm, left column) and mean differences (1985-2014) between the LWAS and NOLWAS simulations (LWAS minus NOLWAS)"*

Figure 4: the symbols rlds and rlus have never been defined.

*rlds and rlus are now defined in the figure caption.*

Figure 5: The colours in the second panel are not very clear and it is difficult to discern the various lines. It is also not very clear what it is meant by "No significant changes in confidence intervals indicated in light color". Is the light colour shading indicating the confidence interval or the significance of the differences between the two runs? How is this computed? Please clarify here or in the mean methods section.

*The colours of the second panel have been changed. The dashed lines of clear-sky lines have also been modified for greater legibility. The light colour shading indicates the confidence interval and it is calculated as follows: 1.96\*standard deviation/square root of number of years.*

Figure A3: it should be "same as Figure 4"

Done.

[Figure]

**Figure 1.** Mean differences (1985-2014) between the LWAS and NOLWAS simulations (LWAS minus NOLWAS) in vertical velocity (wap, 500 and 925 hPa, top left). Vertical profiles of vertical velocity (top right), temperature (bottom left) and specific humidity (bottom right) over the Sahel in September for the LWAS (red) and NOLWAS (blue) simulations. Difference between these two simulations (LWAS minus NOLWAS) is shown (dashed grey line). Confidence intervals for no significant changes indicated in grey light color (Student's t-test, 0.05 level).

[Figure]

**Figure 2.** LW heating rate (tntrl) vertical profiles over the Sahara region in July and September for the LWAS (red) and NOLWAS (blue) simulations. Difference between these two simulations (LWAS minus NOLWAS) is shown in grey. Confidence intervals for no significant changes indicated in grey light color (Student's t-test, 0.05 level). Associated coarse aerosols concentration vertical profiles are shown on the right.

---

## Author Comment (AC2)

We would like to thank the anonymous referee for his comments mentioning different points listed below. The reviewer's comments are in black, and the answers are in red. New information and explanations in the new version of the article are italicized.

**Anonymous Referee 2**

This paper is an important advance in providing better radiative transfer schemes in climate models. The redo of ARPEGE-Climate with longwave thermal IR scattering for large dust aerosols represents new work and is timely. The authors did a very nice job in presenting overall, with only minor English language issues. The authors can clean up some of the language issues and clarify the issue of monthly climatologies. The other problem is with the diagnostics that remove clouds ('clear sky') from a self-consistent climate model but assume that surface temperatures and water vapor are unchanged. These diagnostics should be removed or justified.

New information on the definition of clear-sky and all-sky diagnostics has been added to section 2.2: *"The various radiative diagnostics provided in this study have been computed in all-sky conditions and in clear-sky ones, as classically done. In clear-sky conditions, only clouds are removed, surface temperatures and water vapor remaining unchanged."*

Besides, we acknowledge the disadvantage of the clear-sky diagnostics is that they do not take into account the effects of water vapor, which is important for LW radiative budget. Therefore, this point has been clearly mentioned and Table 1 has been simplified, the results obtained under clear-sky conditions have been put in appendix.

Overall the paper is well written, an important contribution, and nearly ready to publish.

L35-48: This discussion is important here and should do a better job of helping the reader understand the basic physics of the atmosphere and radiative transfer:

- the 8-12 micron window is so well known because H2O and other gases blanket the longer wavelengths and shorter wavelengths; it has nothing to do with the aerosol properties.

We agree with the reviewer's comment. The following sentence has been modified in the text: *"Both Dufresne et al. (2002) and Sicard et al. (2014) underlined that the LW aerosol RF is maximum at wavelengths between 8 and 13 µm, as expected, while Sicard et al. (2014) indicated that large particles have a non-negligible effect in the 17 to 22 µm range at the TOA."*

- it would be nice to discuss what size aerosols need to be to affect this IR window (> 2 microns?), including a discussion of their Q vs effective radius here.

The impact of aerosol size on LW radiation is discussed in the fourth paragraph of the introduction, where we provide information from several publications. Regarding their Q, dust particles are generally weakly hygroscopic, and their ability to absorb water decreases with increasing size (Kumar et al. 2011).

L119: Since you do not calculate the scattering phase function for the aerosols, you are limited to RRTMG's two-stream scattering, which causes heating biases (at least in the uv-vis) over bright albedo regions (Hsu and Prather, 2021, Assessing uncertainties and approximations in solar heating of the climate system. JAMES, 13, e2020MS002131. doi: 10.1029/2020MS002131). Interestingly enough Hsu & Prather did not do LW scattering but only solar. There is nothing that can be done about the use of 2-stream here alas. The other 4+ stream RT codes are at GFDL and CCC.

Thank you for this interesting information. A sentence has been added to the text: *"The use of two-stream scattering in RRTM could also cause some heating biases in the UV and visible spectrum (Hsu and Prather, 2021).*

L153: I am confused as to whether you used AERONET to evaluate the monthly climatology used in your LWAS calculations or to evaluate the daily TACTIC results used to get the monthly means – that would make more sense, then you can use daylight only model data. Can you please evaluate the issues of diurnal aliasing with your TACTIC model? ARPEGE-Climat is using a climatology – right? Also, is the monthly climatology different for each year?

In this article, AERONET data are used to evaluate the 3D monthly climatology of aerosol concentrations used in the LWAS and NOLWAS simulations. The 10-year simulation using TACTIC, on which this climatology is based, was also evaluated with AERONET data in a previous article (Drugé et al. 2022). The issues of diurnal aliasing with our TACTIC model were notably mentioned in our previous article (Drugé et al. 2022). The climatology used here is the same for all years, but evolves monthly over the course of the year.

L162-165: This algorithm is a bit confusing, please make it clear what was done.

In order to have the most accurate comparison with the model, we decided to keep only AERONET monthly data with at least eight daily values to derive the mean of each month, and for a given month we keep only the stations with at least 3 monthly values over the 2000-2020 period. This data selection process has been clarified in the text.

L172. Since dust, esp. large dust, is non-spherical, please comment on the errors resulting form Mie assumption as opposed to using T-matrix or PingYang's ice crystal codes to get the scattering phase function.

Several studies, such as Bellouin et al. 2004 or Colarco et al. 2014, have shown that considering dusts as spherical in climate models has little impact on the DRE at the TOA. This information is given at the end of the fourth paragraph of the introduction.

L173: Glad to see new work on the RI of aerosols.

L218: I thought that the ARPEGE-Climat model being used here for LW scattering is just using an aerosol pre-calculated climatology – what does this r value mean – it should be daily, not monthly means I would expect. If you are evaluating the TACTIC run to generate the climatology, then this discussion needs to be revised.

Correlation coefficients are calculated here to assess the monthly AOD simulated by the ARPEGE-Climat model, which effectively uses a 3D monthly climatology of aerosol concentrations, against the monthly climatology of AERONET AOD data.

L238: "several meteorological fields" You have only one or two meteorological fields (with and without LWAS). Maybe you mean diagnostics? or quantities?

Yes, we're talking about diagnostics here.

L243: These 4 months "cover" or "span" the period of max AOD, they do not "correspond" to it – the latter implies that they are the four months of max AOD.

Done. The sentence has been modified.

L245: "clear-sky" conditions are artificial and only in a model, what is the purpose here? You should focus on "all-sky" conditions. Also you need the full climate response in ARPEGE since the surface T is changing. This discussion is odd – who cares if it is more significant in clear-sky? We are talking climate and monthly mean aerosols!

We have reoriented slightly our study which now focuses mainly on results under "all-sky" conditions. We have simplified our Table 1 but we kept the "clear-sky" section in Appendix as our results under "clear-sky" conditions show that the LW scattering of coarse aerosols has a direct impact on radiation. Furthermore, this distinction between "all-sky" and "clear-sky"

is also often used in the scientific literature, so we thought it would be useful to provide this information, at least in the appendix.

100     L255: Here is the climate response – excellent.

L260: Very interesting climatic shift due to the LWAS, how significant is the change in high cloud (Fig 5) or convective rain – I did not notice any discussion of ensembles or climate variability?

105     In this study, the significance of changes between the two simulations (LWAS and NOLWAS) is tested using a Student's t-test (0.05 level). Indeed, a larger study, with several climate models or an ensemble of simulations would be very useful and interesting to support or not the different results of this study.

L304: How robust are these differences to issues of climate variability, and would having an interactive aerosol calculation
110     (vs. monthly mean) change these results? e.g. large dust aerosol loading depends on wind shear, lack of cirrus, etc? Also the wind/rain biases in L334.

This study is based on the analysis of simulations carried out over a period of 30 years, which makes it possible to obtain robust and significant results despite climate variability. Having an interactive aerosol calculation could modify these results.
115     Indeed, as these results are obtained with aerosol AOD averaged over the month, AOD peaks are not taken into account. Use an interactive aerosol scheme would make it possible to study the impact of aerosol scattering in the LW spectrum during strong wind events generating high dust AOD loads, or during heat waves, etc.

Correcting for wind and rain biases in the model would also provide a better estimate of the impact of aerosol LW scattering. In
120     the northern part of the region studied in this study, a correction of these biases would result in a less strong AOD and therefore potentially smaller impacts and vice versa in the southernmost part of this region. It is important to note, however, that the strongest effects are not always co-localized with the strongest AODs.

L313: Does ecRad actually do 3D RT? (pardon my ignorance here)
125

The 3D effects of clouds are not taken into account in our simulations for reasons of computing cost. However, this option is available in ecRad with the SPARTACUS solver.

Table 1. The extensive comparison of All-sky vs Clear-sky is becoming too artificial. For SW, this is fine because water vapor
130     does not affect much. Clouds are clearly correlated with the water vapor and hence just removing clouds without including their effect on water is misleading.

Table 1 has been simplified and the results obtained under clear-sky conditions have been put in Appendix. Moreover, new information and limitations on clear-sky and all-sky diagnostics have been added to section 2.2.
135

Figure 3. A lot of the colored signal here appears to be statistically not significant w.r.t. climate noise. There are some hatched areas, but most of your square are not. For Figure 4, the radiation is clearly significant, but the T is not.

In fact, not all diagnostics are affected in the same way by taking into account the LW diffusion of aerosols. These differences
140     are discussed in the article. In addition, other diagnostics (LW heating rate, vertical velocity, temperature and specific humidity vertical profiles) have also been added and discussed in this study.

One odd question: does you model include LW scattering by cirrus and stratus? Can you comment on that impact (which would seem to be bigger issue than dust)?
145

Yes, the LW scattering by clouds is taken into account in our model. Cloud longwave scattering is frequently omitted in the radiation schemes of atmospheric models, even if it can increase longwave cloud radiative effect by around 10% globally (Costa and Shine, 2006).

150   Please finish the Dufresne 2002 reference by adding: 'doi:10.1175/1520-0469(2002)059<1959:LSEOMA>2.0.CO;21, 2002'

Done.

---

## Referee Report (RR1)

Review of Drugé et al. "Radiative and climate effects of aerosol scattering in long-wave radiation based on global climate modelling".

I have reviewed this manuscript considering the interactive discussion that preceded this revised manuscript. The study provides an interesting insight into the role of longwave scattering from aerosols and its representation in modelling studies. The introduction very nicely sets up the knowledge gap that the study then focuses on. The methodology is appropriate, and the results provide some clear conclusions. However, I believe there are several elements that need to be discussed further and some limitations to the study that should be included in the conclusions. Therefore, I recommend publication in ACP following some minor revisions that are detailed below.

1. Missing larger dust particles

In the introduction, the authors note that the community has demonstrated an important (yet uncertain) source of sensitivity from coarse dust particles over 20 um in diameter. Di Biagio et al. (2020) make an important point that a lot of the impact from representing these larger dust size modes is due to a compensating reduction in the concentration of the smaller particles that have an opposing radiative effect (cooling vs warming). However, line 211 in the revised manuscript states that the aerosol scheme used to prescribe fields of aerosol extinction only represents dust up to 20 um. If this is the case, then I am interested to know how the authors think their results and conclusions are affected by the omission of larger dust particles. I suggest the authors include a short paragraph in the conclusions section to discuss this in reference to the cited studies from the introduction.

2. Dust evaluation

The authors demonstrate that there is a latitudinal dipole in the coarse AOD bias (around lines 227- 230). Does this point to structural deficiencies in the model? If the model is overestimating coarse dust over the Sahel, then does this weaken their conclusions over the region? I suggest that the authors expand the lines stated to provide a potential explanation for the opposing biases in the region. Do the AeroCom models also demonstrate a dipole in the bias over Northern Africa? This discussion should also be included in the conclusions around line 336, especially with regards to how this influences the other conclusions (i.e., the strong cloud/precip response over the Sahel).

3. Impacts to cloud / precipitation.

I agree with reviewer #2 that there was a lack of in-depth discussion around the drivers of the cloud fraction changes. These changes are key elements of the story. Although the authors have made some progress in this, I believe there are remaining questions. The current explanations are not convincing.

Line 264. What is the mechanism that is driving the enhanced high cloud fraction in all regions of interest? For the Sahel, the authors demonstrate that it is associated with enhanced updraught speeds aloft but do not provide a robust explanation. Is this deep convection? Isn't the atmosphere stabilized? What is happening in the other regions – I suggest the authors include thermodynamic profiles (as A6 for the Sahel) for the other two regions.

Line 279. Where does the significantly enhanced water vapour come from?

On line 279 the authors say there is a reduction in low-level convection due to stabilization but then associate the stabilization with more convective rain below 700 hPa. Please expand this to explain this juxtaposition.

I don't think 'wetter atmospheric layers' adequately explains the precipitation response. This suggests that there is enhanced liquid water content in all clouds throughout the column (do you see this?), but this is not consistent with enhanced convection above 700 hPa (which I assume is deep convection rather than elevated convection?). Looking at the change in precipitation as a function of intensity (mm hr-1) may provide a clearer picture – the lower intensities would be associated with lower altitude clouds / shallow convection and the higher intensities with deep convection.

Finally – have other studies seen cloud responses like this?

4. Conclusions

Line 324. How do the typical treatments compare to this full representation of aerosol LW scattering? Do these results help establish whether they are insufficient?

The authors should consider extending the final paragraph to detail other limitations of the study. This may include (but not limited to...) the lack of dust larger than 20 um, sensitivity to unresolved / parameterized convection, the representation of cloud microphysics, uncertainties in the aerosol model, and remaining uncertainty in the dust refractive indices.

5. Variable names

The variable names are often unintuitive – e.g., tntrl, Wap, rsscs. Please consider replacing all of them with alterative variable names – such as $LW\uparrow_{TOA}$, $LW\downarrow_{SFC}$ clear-sky, $T_{SFC,max}$.

6. Figures

The figures need some work to get to a necessary standard for publication.

Figure 1 is not colorblind friendly and could benefit from thicker lines.

Figure 2 I suggest plotting the filled circles above the coast/country lines and clearly separating the notation for 8 and 10 (is it??). The colorbar labels has been cut off.

Please try to avoid combining plots as in Figure 3 and Figure 4 (and others in the appendix) or have different sized subplots and axis labels etc. I hope you agree it doesn't look great. A6 is another plot that does not look good due to different sized subplots and text.

Consider using thicker lines in all line plots.

For the global/regional plots, consider reducing either the cross hatching or replace with dots – it's almost impossible to see the magnitude of the response below the hatching.

---

## Author Response (AR2)

We would like to thank the anonymous referee for his comments mentioning different points listed below. The reviewer's comments are in black, and the answers are in red. New information and explanations in the new version of the article are italicized.

**Anonymous Referee 1**

I thank the authors for the revised manuscript, which can now be accepted. I just have a minor technical corrections: in Figure 5 the caption still says "No significant changes in confidence intervals indicated in light color" I think the correct line should be "confidence intervals indicated in light color". Please double check the caption.

Done.

We would like to thank the anonymous referee for his comments mentioning different points listed below. The reviewer's comments are in black, and the answers are in red. New information and explanations in the new version of the article are italicized.

**Anonymous Referee 3**

Review of Drugé et al. "Radiative and climate effects of aerosol scattering in long-wave radiation based on global climate modelling".

I have reviewed this manuscript considering the interactive discussion that preceded this revised manuscript. The study provides an interesting insight into the role of longwave scattering from aerosols and its representation in modelling studies. The introduction very nicely sets up the knowledge gap that the study then focuses on. The methodology is appropriate, and the results provide some clear conclusions. However, I believe there are several elements that need to be discussed further and some limitations to the study that should be included in the conclusions. Therefore, I recommend publication in ACP following some minor revisions that are detailed below.

1. Missing larger dust particles
In the introduction, the authors note that the community has demonstrated an important (yet uncertain) source of sensitivity from coarse dust particles over 20 um in diameter. Di Biagio et al. (2020) make an important point that a lot of the impact from representing these larger dust size modes is due to a compensating reduction in the concentration of the smaller particles that have an opposing radiative effect (cooling vs warming). However, line 211 in the revised manuscript states that the aerosol scheme used to prescribe fields of aerosol extinction only represents dust up to 20 um. If this is the case, then I am interested to know how the authors think their results and conclusions are affected by the omission of larger dust particles. I suggest the authors include a short paragraph in the conclusions section to discuss this in reference to the cited studies from the introduction.

We agree this is an important limitation to take into account in this study. Our aerosol scheme has indeed dust aerosols up to 20 um, as in most climate models. We have therefore added a sentence in the conclusion to mention it: *"However, it is important to note that these results may be underestimated because the coarsest dust particles (with a diameter greater than 20 um) are not yet taken into account in the model."*

2. Dust evaluation
The authors demonstrate that there is a latitudinal dipole in the coarse AOD bias (around lines 227- 230). Does this point to structural deficiencies in the model? If the model is overestimating coarse dust over the Sahel, then does this weaken their conclusions over the region? I suggest that the authors expand the lines stated to provide a potential explanation for the opposing biases in the region. Do the AeroCom models also demonstrate a dipole in the bias over Northern Africa? This discussion should also be included in the conclusions around line 336, especially with regards to how this influences the other conclusions (i.e., the strong cloud/precip response over the Sahel).

Our results indeed show a slight overestimation of coarse AOD in Algeria (Tamanrasset) and in Saudi Arabia, as well as an underestimation of coarse AOD at southern latitudes (notably in Niger and Senegal), thus leading to this latitudinal dipole in the coarse AOD bias. This bias may come either from dust emissions, which themselves depend on the wind simulated by the model, or from a bias in precipitation. The simulation used to produce these aerosol fields is already known to suffer from a misrepresentation of the West African monsoon (Roehrig et al. 2020), which could contribute to biases in dust emissions near the Sahel. AeroCom phase 3 models do not show this dipole, but they underestimate coarse AOD by 46 % when compared to 222 AERONET stations around the globe, as already mentioned in the manuscript.

The results presented in this study depend in fact on these different biases in the coarse AOD and we can legitimately assume that in regions where the coarse aerosols AOD is underestimated (as is the case over the Sahel), the radiative and climatic impacts shown are also underestimated. Conversely, the results shown over the Sahara, where the coarse AOD is overestimated, are perhaps less robust. The following two sentences have been added to the conclusion: *"It is important to note, however,*

50  *that these results may be affected by the various coarse AOD biases discussed above."* and *"Moreover, the underestimation of coarse AOD near the Sahel, probably linked to a poor representation of the African monsoon in the simulation in which the AOD dataset was produced (Roehrig et al. 2020), could contribute to underestimating the effects in this region."*

3. Impacts to cloud / precipitation.

55  I agree with reviewer 2 that there was a lack of in-depth discussion around the drivers of the cloud fraction changes. These changes are key elements of the story. Although the authors have made some progress in this, I believe there are remaining questions. The current explanations are not convincing. Line 264. What is the mechanism that is driving the enhanced high cloud fraction in all regions of interest? For the Sahel, the authors demonstrate that it is associated with enhanced updraught speeds aloft but do not provide a robust explanation. Is this deep convection? Isn't the atmosphere stabilized? What is happen-
60  ing in the other regions – I suggest the authors include thermodynamic profiles (as A6 for the Sahel) for the other two regions.

The increase in high clouds observed over the Sahel in September is associated with a deep convection regime (negative vertical velocity absolute value, see Figure A7-A) which is significantly reinforced in the LWAS simulation. A new parameter, wind divergence, has been added to the study (Figure A7-B and Figure A9). Figure A7-B clearly shows that the increase in convec-
65  tion observed above 700 hPa is caused by a significant increase in wind convergence at 700 hPa. The following sentence has been added/modified in the text for greater clarity: *"Conversely, above 700 hPa, Figure A7-A highlights a significant increase in convection. Associated with a deep convection regime (negative vertical velocity absolute value, Figure A7-A) and coupled with a humidity augmentation (Figure A7-C), this stronger convection, caused by a significant increase in wind convergence at 700 hPa (Figure A7-B and Figure A9), favors high clouds over the Sahel in September."*

70
A similar study has now been added over the Sahara in August, where a significant increase in high clouds was also observed. Figure A8, showing vertical profiles of vertical velocity, wind divergence, specific humidity and temperature over the Sahara in August has also been added to the study. As previously over the Sahel, we can see here that the high clouds increase is due to an increase in humidity (Figure A8-C) coupled with a significant increase in convection (Figure A6 and Figure A8-A) due to
75  a significant augmentation in wind convergence at 700 hPa (Figure A8-B). The following sentence has been added to the text of the article: *"Similarly, the increase in high clouds observed over the Sahara in August is also due to an increase in humidity (Figure A8-C) coupled with a significant convection increase (Figure A6 and Figure A8-A) above 700 hPa, which is due to a significant increase in wind convergence at 700 hPa (Figure A8-B)."*

80  This study was not carried out over the Arabian Peninsula because no significant increase in high clouds was observed over this region (Figure 5).

Line 279. Where does the significantly enhanced water vapour come from?

85  The increase in water vapour observed in September over the Sahel is due to an increase in wind from the Atlantic Ocean. Figure A10, which shows a significant increase in ua (eastward wind) over the Sahel in September from 850 hPa, has been added to the article. The following sentence has also been added to part 5 of the article: *"The humidity augmentation observed over the Sahel in September is due to an increase in wind from the Atlantic Ocean, particularly at 850 hPa and above as shown in Figure A10."* The increase in humidity observed over the Sahara in August seems to be due more to an increase in wind
90  from the Sahel, which is wetter. The following sentence has been added to the article: *"The humidity increase observed here over the Sahara in August seems to be due to an increase in wetter winds from the Sahel (not shown here)."*

On line 279 the authors say there is a reduction in low-level convection due to stabilization but then associate the stabilization with more convective rain below 700 hPa. Please expand this to explain this juxtaposition.
95
We acknowledge this sentence was not very clear. The increase in precipitation observed here over the Sahel in September is rather associated with an increase in humidity and in low and high clouds. This has been clarified in the text of the article: *"These increases in humidity and in low and high clouds over the Sahel during September may explain the increase in convec-*

*tive rain previously observed during this month."*

I don't think 'wetter atmospheric layers' adequately explains the precipitation response. This suggests that there is enhanced liquid water content in all clouds throughout the column (do you see this?), but this is not consistent with enhanced convection above 700 hPa (which I assume is deep convection rather than elevated convection?). Looking at the change in precipitation as a function of intensity (mm hr-1) may provide a clearer picture – the lower intensities would be associated with lower altitude clouds / shallow convection and the higher intensities with deep convection. Finally – have other studies seen cloud responses like this?

The output of liquid water content has not been saved, so we cannot analyze the liquid water content in all clouds throughout the column. Similarly, looking at the change in precipitation as a function of intensity would indeed be very interesting to associate the precipitation increase with an augmentation in low clouds or, conversely, with an augmentation in high clouds, but the necessary hourly output of precipitation have not been saved so that we cannot tackle this issue.

Similar results, showing the impact of aerosols on convection, have already been shown in previous studies. The sixth IPCC report, which is based here on the study by Wang et al. 2013, shows that an absorbing aerosol layer can lead to a decrease in convection in the lower layers of the atmosphere below this aerosol layer and, conversely, an increase in convection facilitating more intensive vertical development of clouds in the upper atmospheric layers.

4. Conclusions
Line 324. How do the typical treatments compare to this full representation of aerosol LW scattering? Do these results help establish whether they are insufficient?

This is indeed an interesting question. However, to answer it, it would have been necessary to carry out a simulation similar to the LWAS simulation but with a simplified account (corrective factor) of aerosol scattering in the LW spectrum and to compare the results. This question could be addressed in another study.

The authors should consider extending the final paragraph to detail other limitations of the study. This may include (but not limited to...) the lack of dust larger than 20 um, sensitivity to unresolved / parameterized convection, the representation of cloud microphysics, uncertainties in the aerosol model, and remaining uncertainty in the dust refractive indices.

The concluding paragraph on the limitations of the study has been amended.

5. Variable names
The variable names are often unintuitive – e.g., tntrl, Wap, rsscs. Please consider replacing all of them with alterative variable names – such as LW TOA, LW¯SFC clear-sky, TSFC,max.

The text has been simplified to include fewer abbreviations for variable names. The following acronyms have been removed from the text and only retained in the figures: clh, cll, rls, rlut, tasmin, wap, tntrl.

6. Figures

The figures need some work to get to a necessary standard for publication.

Figure 1 is not colorblind friendly and could benefit from thicker lines.

The yellow lines have been changed to brown and the lines have been thickened.

Figure 2 I suggest plotting the filled circles above the coast/country lines and clearly separating the notation for 8 and 10 (is it??). The colorbar labels has been cut off.

150 Filled circles have been placed above the coast/country lines. Numbers 8 and 10 have also been clearly separated and the cut colorbar labels have been corrected.

Please try to avoid combining plots as in Figure 3 and Figure 4 (and others in the appendix) or have different sized subplots and axis labels etc. I hope you agree it doesn't look great. A6 is another plot that does not look good due to different sized
155 subplots and text.

For the sake of clarity, and in order to have as much information as possible on the same figure, we thought it wise to keep these different figures. However, subplots, axis labels, etc have been harmonized to produce clearer figures. Figure A6 has also been split into two figures (vertical profiles have been given a second figure).
160
Consider using thicker lines in all line plots.

Done.

165 For the global/regional plots, consider reducing either the cross hatching or replace with dots – it's almost impossible to see the magnitude of the response below the hatching.

Done, the cross hatching has been reduced to view data more clearly.

---

## Author Response (AR3)

We would like to thank the anonymous referee for his comments mentioning different points listed below. The reviewer's comments are in black, and the answers are in red. New information and explanations in the new version of the article are italicized.

**Anonymous Referee 3**

5

The authors have taken all of my comments into consideration and have either provided acceptable responses or acceptable revisions to the manuscript. I recommend this manuscript is published as is. I did note a typo in Figure A7-A with 'vertical velociry'. I presume this can be corrected during the publication process.

10   The correction has now been made.

We would like to thank the anonymous referee for his comments mentioning different points listed below. The reviewer's comments are in black, and the answers are in red. New information and explanations in the new version of the article are italicized.

**Anonymous Referee 2**

The further revised manuscript does dig deeper in the relationship between longwave aerosol scattering and the feedback observed with the other model variables, though the full explanation of the model changes remains tricky.

Some of the new sections are somewhat speculative regarding particular feedbacks and I feel they confuse a bit more the explanation of the results. For example:

lines 278-285: I still find puzzling the coincident stabilization of the lower layers together with an increased deep convection above 700 hPa in the same region. Lines 283-285 are speculative, and I find to not help clarifying the whole situation.

line 289-290: again, quite speculative especially because the actual anomaly is not shown. If the observation is not clear, I don't think it should be mentioned if it does not help the explanation.

Indeed, these sentences can complicate the results. We have therefore decided to remove them to clarify the text. We have also moderated the conclusions, and explained that these are only hypotheses to explain the changes in clouds and precipitation caused by the adding of longwave scattering by aerosols.

The various changes are highlighted in the text of the article. The following sentence has been added to the results section: *"To summarize, it would appear that the addition of aerosol diffusion in the LW contributed to opposite changes in the lower layers (reduced convection and increased humidity) and the middle and upper troposphere (increased convection and clouds) in the Sahel in September, with a potential important role of wind."*. Another sentence has also been added to the conclusion: *"These decreases in surface temperature are associated with changes in clouds, wind circulation and atmospheric stability, varying from month to month and from region to region."*

I think the uncertainty of the links connecting the activation of longwave aerosol scattering and changes in the model should be acknowledged more clearly in the conclusion, together with the limitations already discussed. In particular, some of the observed effects might be model-specific.

To take account of these uncertainties, the following sentence has been added to the conclusion: *"The results presented here may also be model-dependent, and further studies using different climate models would be valuable to assess the robustness of these findings."*